# Projected changes to extreme freezing precipitation and design ice loads over North America based on a large ensemble of Canadian regional climate model simulations

Dae Il Jeong[1], Alex J. Cannon[2], Xuebin Zhang[1]

[1] Climate Research Division, Environment and Climate Change Canada, Toronto, Ontario, M3H 5T4, Canada
[2] Climate Research Division, Environment and Climate Change Canada, Victoria, British Columbia, V8W 2Y2, Canada

*Correspondence to*: Dae Il Jeong (daeil.jeong@canada.ca)

**Abstract.** Atmospheric ice accretion caused by freezing precipitation (FP) can lead to severe damage and failure of buildings and infrastructure. This study investigates projected changes to extreme ice loads – those used to design infrastructure over North America (NA) – for future periods of specified global mean temperature change (GMTC), relative to a recent 1986-2016 period, using a large 50 member initial condition ensemble of the CanRCM4 regional climate model driven by CanESM2 under the RCP8.5 scenario. The analysis is based on three-hourly ice accretions on horizontal, vertical, and radial surfaces calculated based on FP diagnosed by the offline Bourgouin algorithm as well as wind speed during FP. The CanRCM4 ensemble projects an increase in future design ice loads for most of northern NA and decreases for most of southern NA and some northeastern coastal regions. These changes are mainly caused by regional increases in future upper level and surface temperatures associated with global warming. Projected changes in design ice thickness are also affected by changes in future precipitation intensity and surface wind speed. Changes in upper level and surface temperature conditions for FP occurrence in CanRCM4 are in broad agreement with those from nine global climate models, but display regional differences under the same level of global warming, indicating that a larger multi-model, multi-scenario ensemble may be needed to better account for additional sources of structural and scenario uncertainty. Increases in ice accretion for latitudes higher than 40°N are substantial and would have clear implications for future building and infrastructure design.

## 1 Introduction

Atmospheric ice accretion caused by ice storms is a frequent and major natural disturbance that can affect wind energy generation (Yang et al., 2015), urban functioning (Hauer et al., 2011; Armenakis and Nirupama, 2014), communication structures (Mulherin, 1998), the forestry sector (Proulx and Greene, 2001; Seidl et al., 2017), and electrical infrastructure (Fu et al., 2006; Rezaei et al., 2016; Jeong et al., 2018). For example, freezing precipitation that fell during the ice storm in early January 1998 over eastern Canada and the northeastern United States (US) claimed about 47 lives and brought down 1,000 transmission towers, 30,000 utility poles, and millions of trees, and caused disruptions to farms and transportation systems (Henson et al., 2007). This hazard eventually led to the failure of electricity supply to 3.5 million people, caused US $4 billion

damages, and resulted in more than 840,000 insurance claims in Canada and the US (Cortinas, 2000; Lecomte et al., 1998). The ice storm that occurred in late December 2013 over eastern Canada brought down about 500 transmission wires, which led to power outages for more than one million people, and caused substantial damage to plants, urban trees, vehicles, and buildings (Armenakis and Nirupama, 2014). This ice storm resulted in over CAD $106 million in damages to the city of

Toronto and around $200 million of insurance losses (City Report, 2014). Given these significant effects, atmospheric ice thickness caused by freezing precipitation (e.g., on horizontal, vertical, or radial surfaces) is a major environmental load that is incorporated into building and infrastructure design standards over North America (e.g., Canadian Standard Association (CSA), 2010; 2013; 2014; American Society of Civil Engineers (ASCE), 2006).

        Several conceptual models have been proposed to estimate historical amounts of ice accretion on exposed surfaces

from ice storms using meteorological weather observations such as amount and duration of freezing precipitation, surface wind speed, and air temperatures (e.g., Chaîné and Castonguay, 1974; Yip, 1995; Jones, 1998; Makkonen, 1998). These models have provided estimates of design ice loads and risk assessments for overhead transmission lines (e.g., Lamraoui et al., 2013; Nygaard et al., 2014; Zhu et al., 2014) and tower structures (e.g., Mulherin, 1998). In particular, maps of design values for ice thickness (i.e., 50-year return level) by Chaîné (Chaîné and Skeates, 1974) and CRREL (Cold Regions Research and

Engineering Laboratory) (Jones, 1998) have been provided for design standards in Canada (CSA, 2010; 2013; 2014) and United States (US) (ASCE, 2006), respectively.

        Historically, design standards have assumed a stationary climate. However, global climate models (GCMs) project an increase in global mean temperature of 4.8°C from pre-industrial conditions by the end of the 21st century if the growth of greenhouse gas emissions follows a "business as usual" pathway (Inter-Governmental Panel on Climate Change (IPCC), 2013).

Warmer temperatures could directly affect the frequency of ice storms, which generally occur when surface temperatures are within the range -10~0°C (Cortinas et al., 2004). Additionally, freezing precipitation and attendant ice accretion respond to multiple other driving variables, including precipitation, surface wind, and the vertical temperature profile, all of which may change in a warming climate. Therefore, it is important to evaluate how extreme ice loads will change in the future, as these extremes will have significant implications for existing facilities designed based on historical annual maximum ice thickness

data as well as future infrastructure design (Jeong et al., 2018).

        However, few studies have focused on projected changes to extreme ice loads and associated impacts on designing buildings and infrastructure. Recently, Jeong et al. (2018) quantified changes to design ice loads for overhead transmission lines for two future periods (i.e., 2041-2070 and 2071-2100) over Canada relative to 1978-2005 period, using three regional climate change simulations obtained from fifth generation Canadian Regional Climate Model (CRCM5); two simulations

driven by Canadian Earth System Model 2 (CanESM2; Arora et al., 2011) under Representative Concentration Pathways (RCP) 4.5 and 8.5 scenarios and one simulation driven by Max-Planck-Institut Earth System Model (MPI-ESM; Giorgetta et al., 2013) under RCP 4.5. They found a robust signal in projected changes to design ice thickness across Canada, including significant increases for eastern Canada, scattered increases for south central and western Canada, and some decreases for the east coast and Great Lakes regions. However, results were highly dependent on changes of air temperature caused by the future

emission scenario as well as sensitivity of the driving GCM to the emission scenario. Furthermore, the relative influence of internal variability, which can be substantial for cold season climate processes, especially for near-term projections (e.g., Fyfe et al., 2017), could not be quantified. Therefore, they highlighted the necessity of larger ensembles to evaluate uncertainties in future projections and identify robust climate change signals.

5       The compound nature of the weather hazard, involving multiple drivers operating on small spatiotemporal scales, means that realistic simulation of future climate variables at both fine spatial and temporal resolution are needed to quantitatively assess climate change impacts on design ice loads (Rezaei et al., 2016). Regional climate models (RCMs) are the primary tools used to add physical and spatial detail to GCM projections. RCMs have been employed for regional-scale studies to analyze climate change impacts on different extremes and natural hazards. In particular, Canadian RCMs have been

used for the assessment of extreme precipitation (Diaconescu et al., 2016), temperature (Whan et al., 2016; Jeong et al, 2016a & 2016b), wind & snow (Jeong and Sushama, 2018), floods (Jeong et al., 2014b; Teufel et al., 2017), and droughts (Jeong et al., 2014a), over North America. Although, freezing precipitation is a key variable leading to atmospheric ice accretion, it is rarely available from RCM outputs due to the absence of precipitation typing algorithms in some RCMs, low demand, and limited data storage capacity. Several algorithms have been proposed to diagnose different precipitation types (e.g., snow, ice

pellets, freezing precipitation, and rainfall) or mixtures of these types, applicable for both online and offline calculation using surface and upper level temperatures during precipitation (e.g., Baldwin and Contorno, 1993; Bourgouin, 2000; Matte et al., 2018). The Bourgouin (2000) and Baldwin and Contorno (1993) algorithms are employed operationally in the numerical weather prediction models of Canada and the US, respectively.

       This study investigates projected changes to extreme ice accretion/loads on horizontal, vertical, and radial surfaces

used in the design of buildings and infrastructure across North America for future periods. Given the large uncertainties reported by Jeong et al. (2018), this study attempts to quantify the role of internal variability in the projected changes, using a large 50 member initial condition ensemble of simulations by the Canadian Centre for Climate Modelling and Analysis (CCCma) RCM (CanRCM4) (Scinocca et al., 2016), driven by Canadian Earth System Model 2 (CanESM2) global model (Arora et al., 2011) under historical and RCP8.5 scenarios. Freezing precipitation is calculated offline using the Bourgouin

algorithm (Bourgouin, 2000), as it is not available in CanRCM4. Projected changes to 50-year return level (design) ice thickness are provided at specific levels of future global mean temperature change (GMTC) (e.g., +1, +2, and +3°C) above the baseline 1986-2016 level along with the pre-industrial (PI) 1850-1900 level. Finally, an analysis of RCM-projected changes in each of the relevant driving variables – upper level and surface temperatures, precipitation intensity, and surface wind speed – is used to gain insight into the relative influence of their changes on future freezing precipitation and design ice loads. Model

uncertainty is investigated by analyzing regional changes in sub-daily vertical temperature profiles and surface temperatures in an ensemble of 9 GCM simulations.

## 2 Models and datasets

### 2.1 CanRCM4 simulations

CanRCM4 is developed by CCCma of Environment and Climate Change Canada (ECCC). It has the same dynamical core as the Global Environmental Multiscale (GEM) model, which is an integrated numerical weather forecasting and data assimilation

system developed by ECCC (Côté et al. 1998). CanRCM4 shares the same package of physical parameterizations with the fourth-generation Canadian Atmospheric global climate model (CanAM4) of CCCma (von Salzen et al. 2013), which forms the atmospheric component of CanESM2. In particular, CanRCM4 uses the deep-convection scheme of Zhang and McFarlane (1995), the shallow-convection scheme following von Salzen et al. (2005), and the Canadian Land Surface Scheme version 2.7 (Verseghy, 2012). Scinocca et al. (2016) provides further details for main characteristics and physical parameterizations

of this RCM and its relationship with its parent global model CanESM2.

The experimental domain of CanRCM4 covers North America (NA) and adjoining oceans at 0.44° resolution. This study uses an ensemble of 50 initial condition simulations (Fyfe et al, 2017) driven by CanESM2 (Arora et al., 2011) at the lateral boundaries under the historical (1950-2005) and RCP8.5 (2006-2100) scenarios. Simulations share the same forcings, differing only in initial conditions specified at the beginning of the historical simulation period. RCP8.5 is a high emission

scenario corresponding to radiative forcing of 8.5 W/m$^2$ by the end of the 21st century compared to pre-industrial values (IPCC, 2013); therefore, is an appropriate scenario for business-as-usual and non-climate policy conditions. Moreover, RCP8.5 has closely matched recent emissions (Sanford et al., 2014). From each simulation, 3-hourly temperatures at the surface (2-m) and three upper levels (i.e., 500, 850, and 1000 hPa), surface air pressure, precipitation, and surface (10-m) wind speed are used to calculate 3-hourly offline freezing precipitation (FP) and ice accretion.

### 2.2 Regional reanalysis and observed datasets

The National Centers for Environmental Prediction (NCEP) North American Regional Reanalysis (NARR) dataset is used to evaluate CanRCM4. NARR is a long-term, consistent, high-resolution (approximately 0.3° grid) reanalysis dataset for the NA domain (Mesinger et al., 2006). An advantage of using this reanalysis is that its gridded output is more comparable to CanRCM4 output representing spatial scales of a grid cell, compared to local point station measurements. NARR uses the

Baldwin and Contorno (also called NCEP) algorithm to diagnose freezing precipitation occurrence. Three-hourly temperatures at surface and 11 upper levels from 200 to 1000 hPa, surface air pressure, precipitation, and surface wind speed are used to evaluate FP and ice accretion of CanRCM4. Reanalysis variables have differing levels of reliability, depending on the relative influence of observed data and the atmospheric model (Kalnay et al., 1996). In general, upper air temperature and surface air pressure are most reliable (type A) as they are strongly influenced by observed data; surface air temperature and wind speed

are less reliable (type B) compared to type A as both observational data and the model have a strong influence on the variables; and precipitation should be used with caution (type C) as it is typically completely determined by the model (Kalnay et al, 1996). Unlike most reanalyses, however, Mesinger et al. (2006) reported that NARR could potentially provide more realistic

precipitation fields as it assimilates observed precipitation to adjust the accumulated convective and gridded precipitation based on the difference between the modelled and observed hourly precipitation. Reliability of NARR precipitation will thus be heavily influenced by the density of assimilated rain gauge information, which is low over Canada.

To evaluate the frequency and amount of freezing precipitation diagnosed from CanRCM4, precipitation and manually observed weather condition datasets are obtained from 588 meteorological stations over NA for 1986-2016 period. In particular, daily freezing precipitation series are prepared from 205 meteorological stations operated by ECCC over Canada and series from 383 observation stations are obtained from the National Climatic Database Center (NCDC) Integrated Surface Database (ISD) (Smith et al., 2011) over the US.

## 2.3 GCM simulations

In addition to CanRCM4 simulations, 3-hourly surface temperature and 6-hourly 850 hPa temperature outputs from 9 GCM simulations are used to investigate the mechanisms driving changes in FP occurrence in the future. The 9 simulations considered in this study are obtained from 6 CMIP5 (Coupled Model Intercomparison Project phase 5) models. This includes

- GFDL-ESM2G: Geophysical Fluid Dynamics Laboratory Earth System Model with Generalized Ocean Layer Dynamics component ESM2G
- GFDL-ESM2M: Geophysical Fluid Dynamics Laboratory Earth System Model with Modular Ocean Model component, ESM2M
- GISS-E2-R: Goddard Institute for Space Studies Model E, coupled with the Russell ocean model
- IPSL-CM5-MR: L'Institut Pierre Simon Laplace Coupled Model Version Five-Medium Resolution
- IPSL-CM5-LR-r1~r4: four simulations (r1~r4) from IPSL-CM5-Low Resolution
- NorESM1-M: Norwegian Earth System Model version 1-Medium resolution.

## 3 Methodology

### 3.1 Freezing precipitation

Three-hourly FP amounts are calculated offline by the Bourgouin algorithm from 50 CanRCM4 simulations using surface, 500, 850, and 1000 hPa temperatures, surface air pressure, and precipitation for the 1986-2100 period. When precipitation occurs, the Bourgouin algorithm calculates the rates of four precipitation types (i.e., snow, ice pellets, freezing precipitation, and rain). FP is diagnosed when precipitation falls through warm layers ($> 0°C$) aloft and then arrives at a sub-freezing near-surface layer. This vertical temperature profile is possible when strong warm air advection at upper levels is accompanied by strong cold air advection near the surface. The Bourgouin algorithm calculates areas of positive and negative energy from all possible pressure levels above the surface, and then determines the rate of FP based on the positive and negative areas and specified thresholds (Bourgouin, 2000).

Before investigating future CanRCM4 projections, FP estimated from the 50 CanRCM4 historical simulations is evaluated for the 1986-2016 baseline period by comparing the annual frequency of FP days, annual total FP amounts, and daily extremes (i.e., 20-year return value estimated by fitting the Gumbel distribution to annual maxima using the method of moments) with those from NARR and 588 stations. Depending on temperature, the Baldwin and Contorno (NCEP) algorithm used by NARR determines the initial phase of precipitation at the highest saturated layer as either ice crystals (< -4°C) or supercooled liquid water ($\geq$ -4°C). FP is diagnosed when supercooled water occurs in combination with a sub-freezing near-surface layer, or when ice crystals occur in combination with temperatures of warm upper layers and cold surface layer that exceed specified thresholds (Baldwin and Contorno, 1993).

As FP amounts in CanRCM4 are diagnosed based on offline calculations from a limited number of vertical levels, it is important to evaluate their robustness. Therefore, a sensitivity analysis is carried out to quantify the impact of different vertical temperature resolutions on FP using NARR outputs, which are provided at higher vertical resolution (at 11 upper levels from 200 to 1000 hPa), and the Bourgouin algorithm. Three selected combinations of vertical levels, including one with the 3 levels offered by CanRCM4, are considered: case [1] includes all 11 NARR levels; case [2] includes a subset of 6 levels (200, 400, 600, 800, 900, and 1000 hPa); and case [3] includes the 3 CanRCM4 levels. As case [1] employs high vertical resolution, it can be assumed to be the same as the online simulation. Therefore, comparison between case [1] and NARR permits an evaluation of differences in FP diagnosed by the Bourgouin and NCEP approaches.

### 3.2 Ice accretion

Three-hourly horizontal, vertical, and radial ice accretions are calculated using the Chaîné approach (Chaîné and Skeates, 1974), which is the operational scheme used by ECCC. Ice accretion on a horizontal surface ($T_h$; in/3hr), such as a road or a flat roof, assumes that all FP accretes as ice when the surface temperature is below freezing (i.e., horizontal ice accretion is the same as the FP amount). Three-hourly ice accretion on a vertical surface ($T_v$; in/3hr), such as the sides of towers and bridges, is calculated based on FP amount and surface wind:

$$T_v = 0.078 \, V \, P^{0.88} \tag{1}$$

where $V$ is the wind speed (mph) and $P$ is the FP rate (in/3hr). Ice accretion on a radial surface ($R$; in/3hr), such as transmission lines or cables, is also calculated based on FP amount and wind speed:

$$\Delta R = -r + \left[ r^2 + \frac{K \, r}{2} (T_h^2 + T_v^2)^{1/2} \right]^{1/2} \tag{2}$$

where $K$ is a correction factor, $\Delta R$ is the incremental change in ice thickness, r (in) is the radius of the cylinder, including ice already on the conductor, at the start of the hour. Ice accretion continues until the end of the ice storm event either the surface temperature increases to above 1°C or 7 days have passed without FP (CSA, 2010).

For NARR and CanRCM4 simulations, annual maximum ice thickness time series are prepared for all land grid points over NA. Fifty-year return level (RL) (also referred to as design or extreme) ice thicknesses are estimated by fitting the Gumbel distribution – the extreme value distribution used operationally by ECCC for extreme ice thickness analysis in Canada (CSA,

2010) – to the annual maxima. Projected changes are provided for 31-year future periods at specific levels (e.g., +1, +2, and +3°C) of GMTC above the baseline (1986-2016) as well as the PI (1850-1900) temperatures. The statistical significance of changes in design ice thickness in terms of internal variability is assessed by applying two sample t-test at the 10% two-sided significance level between 50 members of baseline and future period values.

## 3.3 Sensitivity to driving variables

FP storms that lead to extreme ice accretion are compound events with multiple physical drivers. The variables driving changes in FP and associated design ice thicknesses in CanRCM4 are investigated by comparing frequencies of 850 hPa temperatures (T850) greater than 3°C and surface temperature (Ts) within the range of -10~0 °C as well as precipitation occurrence for the freezing season (i.e., from onset to offset of Ts < 0 °C) for each year, between the baseline and future +2°C GMTC periods. The sensitivity of FP and ice accretion to changes in driving variables is also investigated by comparing precipitation intensity and surface wind speed for the freezing season between these periods. Finally, to evaluate model uncertainty, changes in frequencies of the upper level and surface temperature conditions are assessed in 9 CMIP5 GCMs.

## 4 Results

### 4.1 Freezing precipitation

When compared against station observations for the 1986-2016 period, the ensemble means of frequency and amount of daily FP are underestimated by CanRCM4, whereas NARR generally overestimates over most of NA (Figures 1a and 1b). Interestingly, CanRCM4 is generally more consistent with in situ data over Canada and NARR over the US. Biases of CanRCM4 (-0.45 day and 1.15 mm) are smaller than those of NARR (3.25 day and 8.02 mm) over Canada for annual FP days and amounts. The reverse is true over the US; biases of NARR (0.16 day and 2.12 mm) are smaller than those of CanRCM4 (-1.47 day and -3.04 mm). The large bias of NARR over Canada is supported by the findings of Mesinger et al. (2006), which reported that the reliability of NARR precipitation is highly related to the density of rain gauges (i.e., the precipitation is less reliable across Canada as fewer observations are assimilated). For annual extremes, despite the fact that FP in CanRCM4 is diagnosed based on only three vertical temperature levels, CanRCM4 yields similar performance to NARR for 20-year RL of daily maximum FP (Fig. 1c). The CanRCM4 ensemble exhibits relatively small 5~95 percentile ranges for most of northeastern NA and parts of southwestern Canada, particularly for annual total amount and extreme FP, compared to ensemble means. However, it yields similar or larger values of the ranges for northwestern, north-central, and southern NA compared to ensemble means, indicating high levels of internal variability in these regions. Large spatial variability in observed annual total and extreme FP are exhibited over Great Lakes and western coastal regions, and CanRCM4 model shows large internal variability (i.e., larger 5~95 percentile ranges than ensemble means) over the regions.

Online and offline results from NARR show that simulated frequencies and amounts of FP are highly dependent on the diagnosis algorithm. Specifically, the online NCEP algorithm (presented in 2nd column of Fig. 1) yields a much higher

frequency of FP occurrence over northeastern regions of NA relative to the offline Bourgouin algorithm using 11 vertical levels (Fig. 2, case [1]), which should behave in a similar manner as the online Bourgouin simulation due to the large number of vertical levels used. The average of relative absolute bias, i.e., percentage of absolute bias to mean value, in annual FP days of the NCEP to Bourgouin algorithms is 55% across NA. This result is consistent with the finding of Mette et al. (2018) in their study region in southeastern Canada. Larger values from the NCEP algorithm are possible because it can simulate FP when all vertical temperatures are below 0°C if supercooled water is diagnosed at the highest saturated layer. Similar differences over the domain are also apparent for the 20-year RL of daily maximum FP; the average of relative absolute bias (systematic bias) of online NCEP algorithm to the case [1] Bourgouin algorithm is 42% (-1.24 mm) across NA, with larger negative bias (smaller than -3 mm in many grids) over eastern and western Canada.

The impact of vertical temperature resolution on diagnosed FP is summarized in Fig. 2. When compared to the diagnosis based on all 11 vertical levels (case [1]), using three vertical levels (case [3]) generally displays larger differences in annual FP days than using six vertical levels (case [2]) over most of northern NA. However, FP occurrence is generally less sensitive to differences in vertical temperature resolution compared to the choice of diagnosis algorithm. Averages of relative absolute bias between cases [2] and [1] and between cases [3] and [1] are 13% and 20%, respectively. Moreover, extreme FP amounts are less sensitive to vertical temperature resolution than frequency of FP; averages of relative absolute bias in 20-year RL between cases [2] and [1] and between cases [3] and [1] are only 9 and 14 %, respectively. The three cases yield similar levels of performance to each other but perform slightly better than NARR in terms of RMSE when they are compared to site observations, as the Bourgouin algorithm yields much better performance over Canada but slightly worse performance over US, compared to the NCEP algorithm. It is notable that FP types cannot be diagnosed over western high altitude regions in case [3] with three vertical levels as surface pressures are usually lower than 1000 hPa; this is also visible in the CanRCM4 simulations (Fig. 1), which use the same levels.

Investigation of vertical temperature profiles during heavy and light FP events over selected eastern and western regions (Fig. 3) provides some clues as to why a limited number of vertical levels (500, 850, 1000 hPa) leads to relatively large underestimation of FP frequency but good performance for extreme FP amounts. Strongly nonlinear temperature profiles with the requisite warm layer are evident for both heavy and light FP events. When only a few vertical levels are considered, as in CanRCM4, the warm layer can be underestimated, which results in misdiagnosis of FP as ice pellets. However, the warm layer temperature maximum occurs at around 850 hPa, which indicates that the strength of warm air advection leading to FP can generally be resolved. Furthermore, over the eastern region, heavy FP events (> 10 mm/3-hour) are associated with a deeper warm layer than light FP events (< 0.2 mm), which means that there is less potential to misdiagnose FP as ice pellets.

**4.2 Ice accretion**

Spatial distributions of design ice thickness (50-year RL) on horizontal, vertical, and radial surfaces for NARR and the historical CanRCM4 large ensemble simulations (ensemble mean and 5[th] & 95[th] percentiles) are presented in Fig. 4. As explained above, 50-year RL of annual maximum horizontal ice thickness is identical to that for annual maximum daily FP

accumulation. Therefore, spatial patterns for horizontal values are generally consistent with those for extreme FP presented in Fig. 1c. These spatial patterns also show a good agreement with those reported for NARR and CRCM5 by Jeong et al. (2018). Furthermore, CanRCM4 shows better agreement with CRCM5 as compared to NARR, particularly over northeastern NA, because the CRCM5 also uses the Bourgouin approach to diagnose FP. CanRCM4 ensemble yields larger 5~95 percentiles ranges for southeastern NA, indicating larger simulated natural variability in that region than elsewhere.

Vertical thickness values are generally larger than horizontal values, indicating that vertical ice extremes are generally associated with strong surface winds during ice storm events. For instance, based on Eq. 1, vertical thickness is larger than horizontal when 10 (20) mm FP with larger than 5.2 (5.6) m/s wind occur. Regionally, CanRCM4 simulates larger values over south-central, north-central, and northeastern NA, where higher wind speeds are reported based on site observations and reanalysis datasets (Pryor et al., 2009; Jeong and Sushama, 2018). NARR shows larger vertical thickness than CanRCM4 over southeastern NA, where an overestimation of the reanalysis in surface wind speed has been reported (Pryor et al., 2009). The CanRCM4 ensemble yields larger 5~95 percentile ranges of vertical extremes in comparison to horizontal extremes, particularly over the north-central and northeastern NA, due to added variability caused by wind speed to the variability originated from FP.

Radial thickness values are larger than horizontal values, but smaller than vertical values. This occurs because radial ice accretion results from the combined effect of horizontal and vertical ice accretion (see Eq. 2). Particularly over Canada, radial extremes show larger values for Atlantic and Hudson Bay coastal regions, as well as northern Quebec, but smaller values for western and central regions. The spatial pattern for CanRCM4 is consistent with the contour maps of historical ice accretion in CSA (2010; 2013), which were developed based on ice accretion values from the Chaîné approach applied to about 200 Canadian meteorological stations for the period 1953-2006. These spatial patterns are also in a good agreement with those of Jeong et al. (2018), which were obtained from the CRCM5 model and the simpler CRREL approach for ice accretion (Jones, 1998). Furthermore, the spatial distribution of CanRCM4 values over the US is consistent with contour maps in ASCE (2006), which were developed using the CRREL approach on data from 330 meteorological stations for the period since 1959.

## 4.3 Future projections

It should be noted before discussing projections that the surface temperature response of CanESM2 is relatively sensitive to radiative forcings; this suggests higher GMTC compared to the other CMIP5 models (Schleussner et al., 2016). CanESM2 also yields larger GMTC for the 1986-2005 period with respect to the PI period (0.79°C), compared to global observations (0.61°C) (IPCC, 2013). As CanRCM4 simulations driven by CanESM2 for the PI period are not available, projected changes associated with different future GMTC levels are calculated with respect to the 1986-2016 baseline period. However, it should be noted that, in terms of CanESM2, the +0.5, +1, +2, and +3 °C GMTC levels relative to the 1986-2016 baseline are the same as +1.5, +2, +3, and +4°C levels relative to the PI period, as the GMTC from the PI to the 1986-2016 baseline period is +1 °C (i.e., an additional +0.21°C warming from the 1986-2005 period used by IPCC, 2013).

CanRCM4 projects significant decreases in future design ice thickness values on horizontal, vertical, and radial surfaces over most of southern NA and some northeastern coastal regions of NA (Fig. 5). The spatial extent and relative magnitude of the decreases tends to increase as GMTC levels increase. However, the CanRCM4 ensemble also projects significant increases in future extreme ice thickness values over northern NA, with the exception of some Atlantic and Hudson Bay coastal regions. When considering changes in relative terms, CanRCM4 projects larger increases in future design ice thickness values for northwestern NA, and smaller increases for northwestern regions of NA. However, it must be noted that absolute changes in the northeast are actually larger than those in the northwest (not shown) due to the differences in baseline amounts in the different regions. These spatial patterns of projected changes of CanRCM4 are broadly consistent with those found for CanESM2 driven CRCM5 simulation over Canada by Jeong et al. (2018), as FPs of both RCMs are driven by the same GCM and precipitation-type diagnose algorithm. Although, robust changing signals in the projected changes to design ice thickness is obtained from the ensemble mean, the CanRCM4 yields large 5-95 percentile ranges to the variable due to the large natural variability of this extreme. It is notable that increases in vertical design ice thickness are larger than horizontal thickness over the northeastern NA, mainly due to increases in future surface wind speed. Gastineau and Soden (2009), Kumar et al. (2015), and Jeong and Sushama (2018) reported modest increases in extreme surface wind speed for those regions by the end of the 21st century based on CMIP3, CMIP5, and CRCM5 ensembles, respectively, due to amplified baroclinicity and decreases in future surface pressure with associated changes in surface pressure gradient.

Zonal averages of projected changes in annual FP amount and 50-year RL radial ice thickness over 10°N latitude bands are given in Fig. 6 for different levels of GMTC relative to the baseline period. Horizontal and vertical values behave in a similar fashion as radial thickness and are not shown. Annual FP amount exhibits decreases (increases) for the 30-40°N and 40-50°N (50-60°N and 60-70°N) latitude bands as GMTC increases. Interestingly, median changes in radial design ice thickness at the 30-40°N and 40-50°N bands increase until the GMTC reaches +1.5°C and +3.0°C levels relative to the baseline (i.e., +2.5°C and +4.0°C levels relative to PI), respectively. Increases in future precipitation intensity and/or surface wind speed could result in increases in future extreme design ice thickness in those latitudinal bands despite overall decreases in FP frequency and annual amount. It is also notable that radial design ice thickness is projected to increase more than the annual FP amount at the 50-60°N latitude band under the same level of warming. Changes in internal variability in annual FP amount and radial design ice thickness with global warming are also shown in the figure. For instance, the two variables show an increase (decrease) in the 5-95 percentiles range at 60-70°N (30-40°N) band as GMTC increases. The 5-95 percentiles ranges of the radial design ice thickness at 30-40°N, 50-60°N, and 60-70°N bands exclude the zero change line when GMTCs are larger than +3.5°C, +1.0°C, and +1.0°C levels relative to the baseline, respectively, indicating statistically significant changes from natural variability at the 10% significance level.

Radial thickness values are larger than horizontal values, but smaller than vertical values. This occurs because radial ice accretion results from the combined effect of horizontal and vertical ice accretion (see Eq. 2). Particularly over Canada, radial extremes show larger values for Atlantic and Hudson Bay coastal regions, as well as northern Quebec, but smaller values for western and central regions. The spatial pattern for CanRCM4 is consistent with the contour maps of historical ice accretion

in CSA (2010; 2013), which were developed based on ice accretion values from the Chaîné approach applied to about 200 Canadian meteorological stations for the period 1953-2006. These spatial patterns are also in a good agreement with those of Jeong et al. (2018), which were obtained from the CRCM5 model and the simpler CRREL approach for ice accretion (Jones, 1998). Furthermore, the spatial distribution of CanRCM4 values over the US is consistent with contour maps in ASCE (2006),

which were developed using the CRREL approach on data from 330 meteorological stations for the period since 1959.

## 4.4 Sensitivity to driving variables

Projected changes to design ice thickness in CanRCM4 are primarily driven by changes in the vertical temperature profile, which, as discussed by Jeong et al. (2018), affects the frequency of FP events. As illustrated in Fig. 3, FP typically occurs when T850 > 3°C coincides with -10°C < Ts < 0°C. CanRCM4 projects a decrease in the future frequency of these temperature

conditions over southern NA, mainly due to warming at the surface (Fig. 7a). In contrast, CanRCM4 projects an increase in the frequency of favorable temperature conditions over northern NA, due to warming at both upper levels and the surface. Despite decreases in the frequency of surface temperature between -10~0°C, some parts of southern Canada and the northern US are projected to experience an increase in FP events due to the increase in frequency of upper level temperature warmer than 3°C (Fig. 7b). Consequently, CanRCM4 projects significant increases (decrease) in FP occurrence hours over northern

(southern) NA (Fig. 7c). In addition, increases in future precipitation intensity over most of NA (Fig. 7d) can accelerate increases (or delay decreases) in FP and associated design ice thicknesses over northern (southern) NA. Finally, as shown in Fig. 4b, increases in surface wind speed over northeastern NA (Fig. 7e) during the freezing season can increase vertical and/or radial extreme ice thicknesses.

For a given level of GMTC, projected CanRCM4 changes in upper level and surface temperature conditions that favor

FP occurrence (i.e., Fig. 7a and 7b) are generally in good agreement with those projected by 9 CMIP5 GCMs (Fig. 8). However, the GCMs do display some regional differences, particularly over northwestern coastal, northeastern, and central regions of NA. In particular, GFDL-ESM2G, GFDL-ESM2M, IPSL-CM5-MR, IPSL-CM4-LR-r2, IPSL-CM4-LR-r4, and NorESM1-M project decreases in the frequency of surface temperature between -10~0°C (Fig. 8a) and associated frequency of favorable conditions for FP occurrence (Fig. 8b) over some regions in northwestern and/or northeastern NA. Note, however, that some

of the regional differences can be attributed to internal variability, rather than structural uncertainty, as the four IPSL-CM5-LR realizations show different patterns of change. As a further indication of structural uncertainty, GCMs have differing levels of climate sensitivity and hence respond differently to the same forcing scenario (e.g., Schleussner et al., 2016). For instance, the most sensitive (IPSL-CM5-LR and -MR) and least sensitive (GISS-E2-R) GCMs reach the +2°C GMTC level of warming in the 2036-2066 and 2070-2100 periods, respectively.

## 5 Summary and discussion

This study investigates projected changes to extreme ice loads used to design buildings and infrastructure for future periods with different levels of global mean temperature change (GMTC) (relative to the baseline 1986-2016 period) using a 50 member initial condition ensemble of CanRCM4 driven by CanESM2 under the RCP8.5 scenario. The large CanRCM4 ensemble allows the range in design extremes reflecting internal variability to be characterized. As freezing precipitation (FP) is estimated in this study using the Bourgouin algorithm with limited vertical temperature information from CanRCM4, the reliability of the diagnosis is further evaluated by comparing with the NCEP algorithm used in the NARR reanalysis. This assessment reveals that (1) the NCEP algorithm yields more frequent FP over northeastern North America (NA) as compared to the Bourgouin algorithm; (2) the Bourgouin algorithm with a limited number of vertical levels can underestimate the frequency of FP occurrence due to underestimation of positive energy area; and (3) upper level warm advection during FP is well represented by temperature at 850 hPa. Therefore, compared to observations and NARR, CanRCM4 with the Bourgouin algorithm underestimates FP frequency, but the magnitudes of the extreme FPs, which are key determinants of extreme ice thickness, are more comparable to the observations despite using limited vertical levels. Consequently, the spatial pattern of 50-year RL radial ice thickness of CanRCM4 is in a good agreement with the design ice thickness contour maps of CSA (2010; 2014) across Canada and ASCE (2006) across US for the baseline period. Vertical design ice thickness is larger than horizontal thickness in regions with high surface wind speed (i.e., north-central and northeastern NA) during the freezing season.

Future projections by the CanRCM4 ensemble suggests decreases in future 50-year RL of horizontal, vertical, and radial ice thicknesses for most southern NA and some northeastern and Hudson Bay coastal regions but increases in these quantities in most northern NA, except for some northeastern and Hudson Bay coastal regions. Increases in upper level and surface temperatures over NA driven by global warming are identified as the main factors leading to changes in FP frequency and associated design ice thickness. But the responses differ in the southern and northern NA with a decrease in the south and an increase in the north. Vertical design ice thickness is projected to increase more for northeastern NA compared to horizontal thickness due to increases in future surface wind speed during the freezing season.

Projected changes to design ice thickness quantified in this study are useful information for the development of climate-resilient design standards, codes and guides for buildings and infrastructure. Caution in designing for ice loads at latitudes higher than 40°N is warranted due to projected increases in extreme ice thickness. Furthermore, northern NA shows an increase in design ice thickness and an increase in surface wind speed. This may have compounding effect to result in an increase in load larger than the increase in ice load or wind load alone. It is thus important to examine changes in future probability of extreme ice loads occurring simultaneously with extreme wind load in that region. Because of the compounding effect, the joint probability is often considered in design standards as a load combination factor (i.e., CSA, 2014).

This study employs a single RCM with one driving GCM under the RCP8.5 emission scenario. The spatial patterns of projected changes to FP and ice thickness from CanRCM4 are largely consistent with those projected with a different regional model (CRCM5) under the same and a different driving GCM over Canada (Jeong et al., 2018). This provides some

confidence about CanRCM4 projected changes. We have attempted to separate projection uncertainty due to internal variability from those due to differences in climate model sensitivity and in emission scenarios by expressing projected changes under different future GMTC levels relative to the baseline period. However, different CMIP5 GCMs do project slightly different patterns of surface and upper level temperature changes at the same GMTC level (e.g., pattern differences between IPSL-

CM5-LR runs in Fig. 8b). This indicates that different models may have slightly different dynamic responses at the same warming level. It is thus important to include other driving GCMs to fully represent uncertainty in the future projection.

**Acknowledgements**

We thank Guilong Li and Robert Morris for providing helpful information and comments on this study. We also would like to thank the anonymous reviewers for their constructive feedback.

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

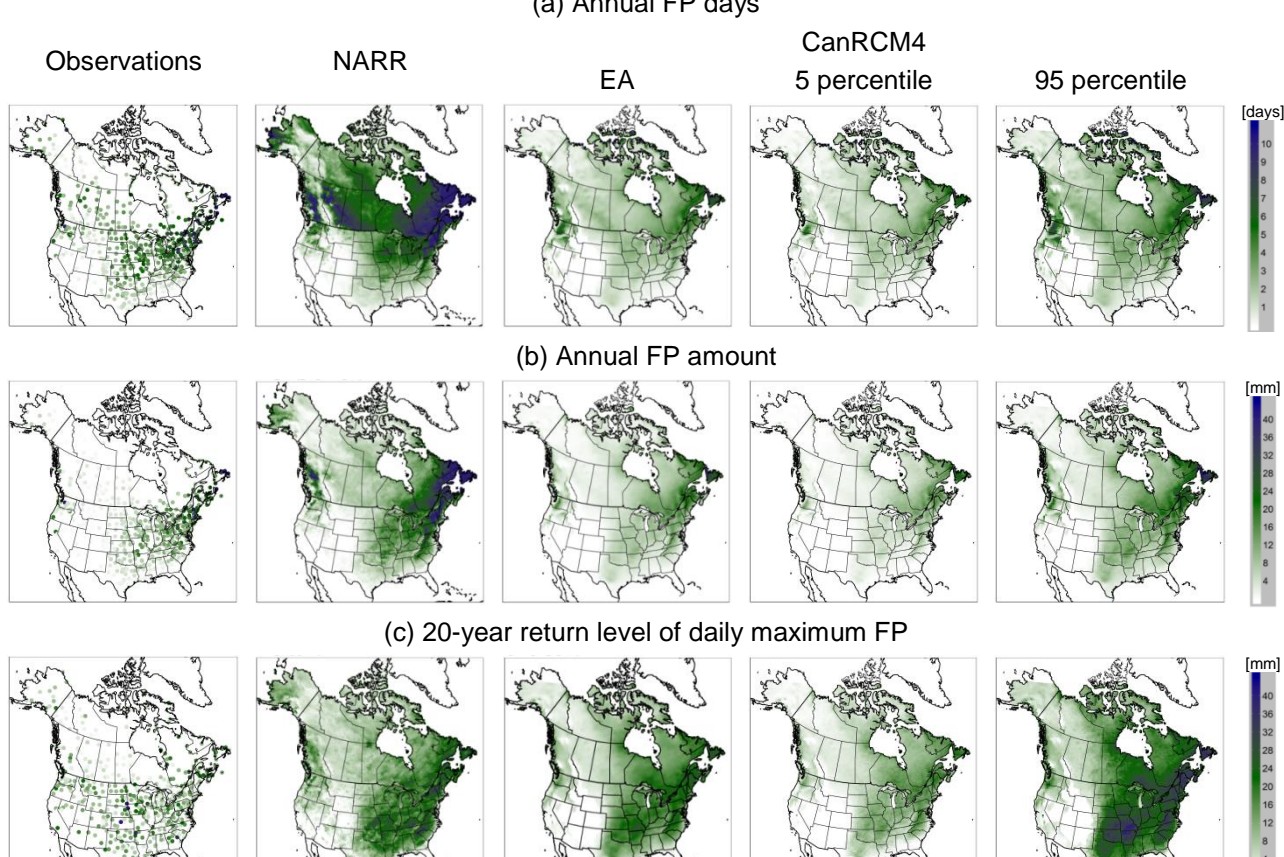

**Figure 1: Mean values of (a) annual freezing precipitation (FP) days and (b) total amount as well as (c) 20-year return levels of daily maximum FP for observations, NARR, and CanRCM4 (ensemble average (EA) and 5 & 95 percentiles) of for the baseline 1986-2016 period.**

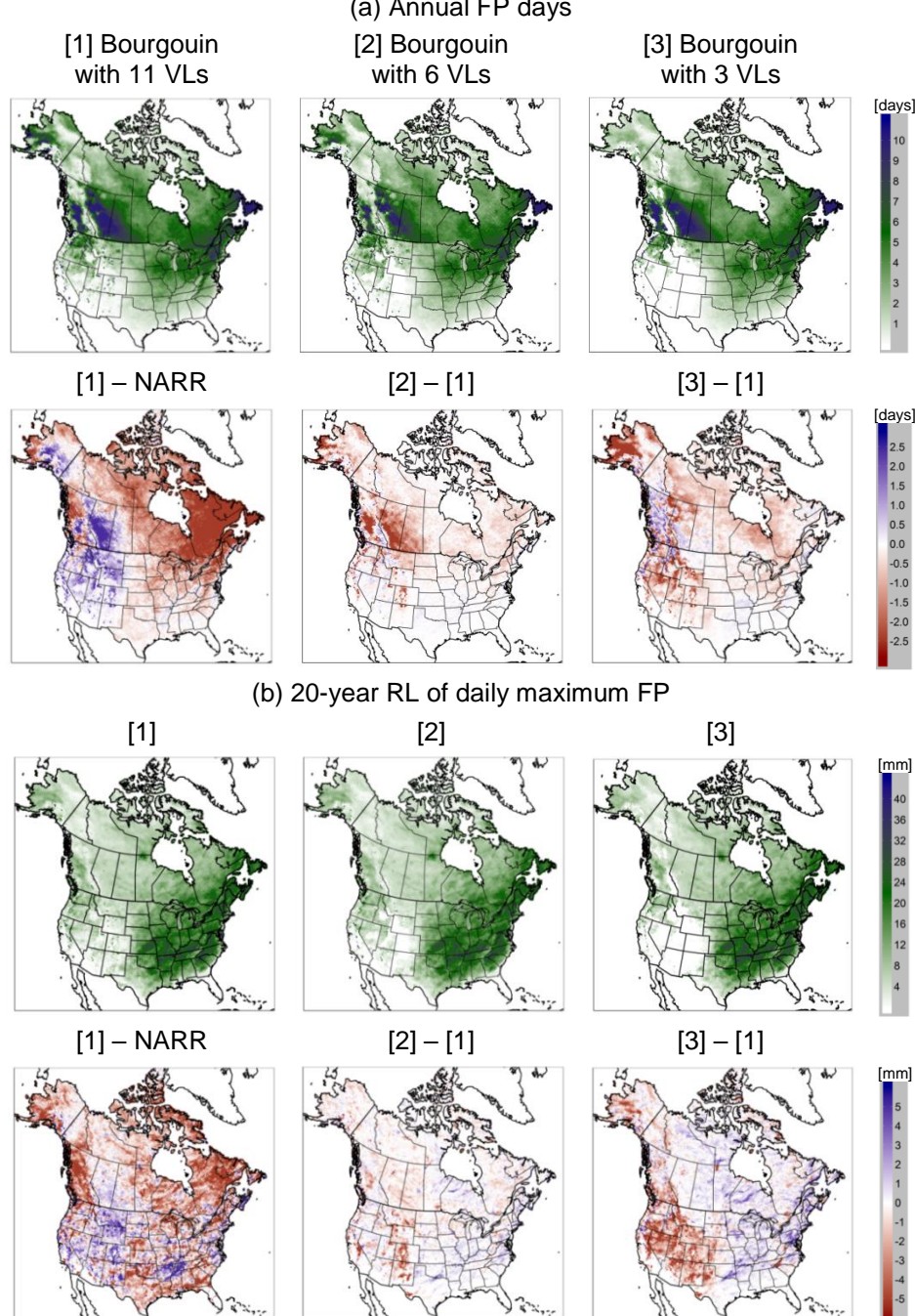

**Figure 2: Annual freezing precipitation (FP) days and 20-year RL of daily maximum FP obtained by offline Bourgouin algorithm with 11 (200, 300, 400, 500, 600, 700, 800, 850, 900, 950, & 1000 hPa; case [1]), 6 (200, 400, 600, 800, 900, & 1000 hPa; case [2]), and 3 (500, 850, & 1000 hPa; case[3]) vertical levels (VLs) of NARR outputs for the baseline period. The last case has identical VLs to CanRCM4. The two statistics from NARR (with online NCEP algorithm) are presented in Fig. 1. Differences between NARR and case [1], cases [1] and [2], and cases [1] and [3] are also provided.**

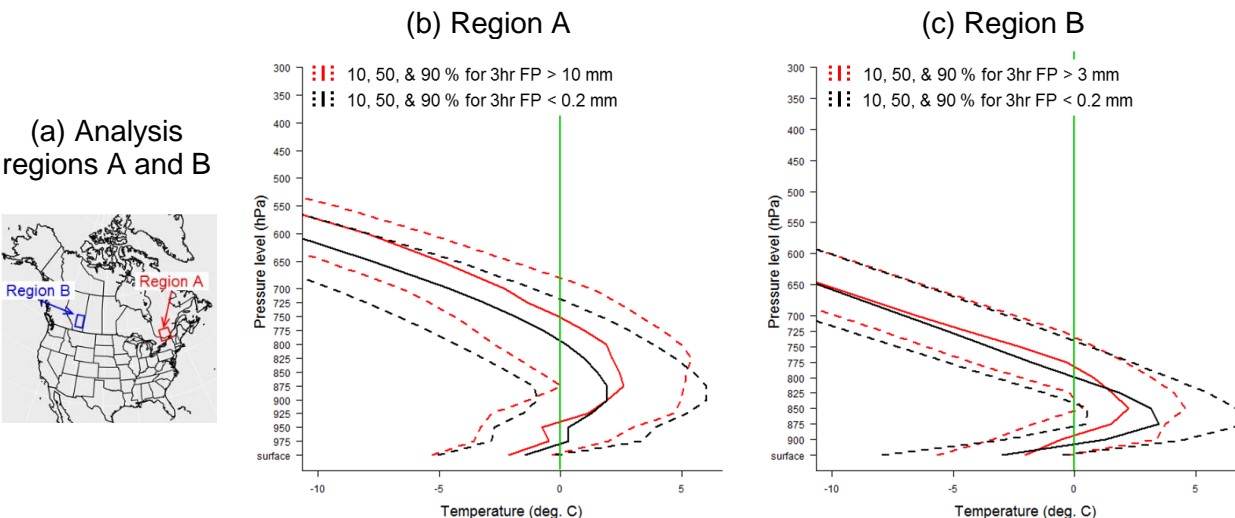

**Figure 3: Distributions (10, 50, and 90 percentiles) of vertical temperature profiles during heavy and light FP obtained by offline Bourgouin algorithm with 11 NARR vertical levels (i.e., case [1] in Fig. 2) at two selected analysis regions located in eastern and western North America for the baseline period. Different thresholds of heavy freezing precipitations (10 and 3 mm) are used for the regions A and B by considering their wet and dry climates.**

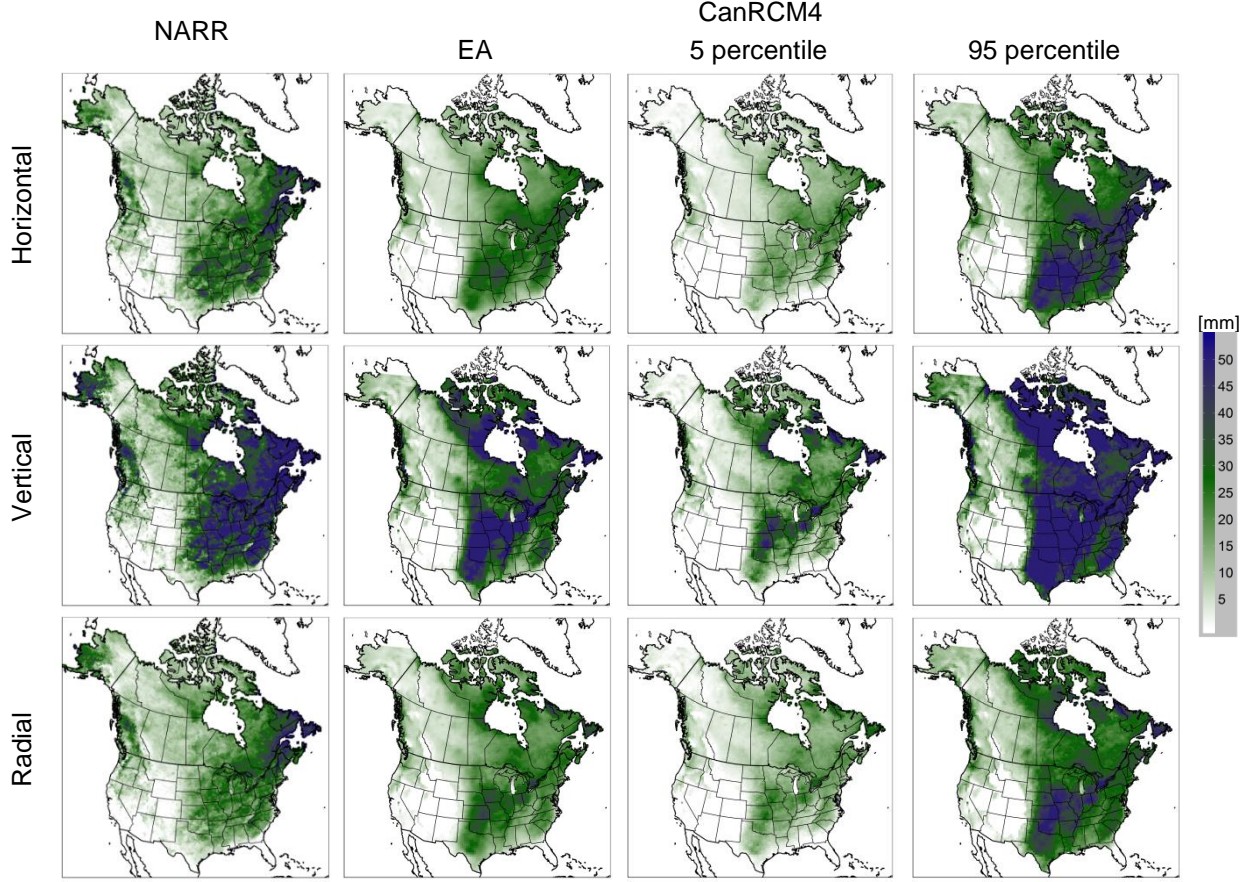

**Figure 4: Fifty-year return level of annual maximum ice thickness on horizontal, vertical, and radial surfaces for NARR (column 1) and CanRCM4 (ensemble average (EA) and 5 & 95 percentiles (columns 2-4)) for the baseline period.**

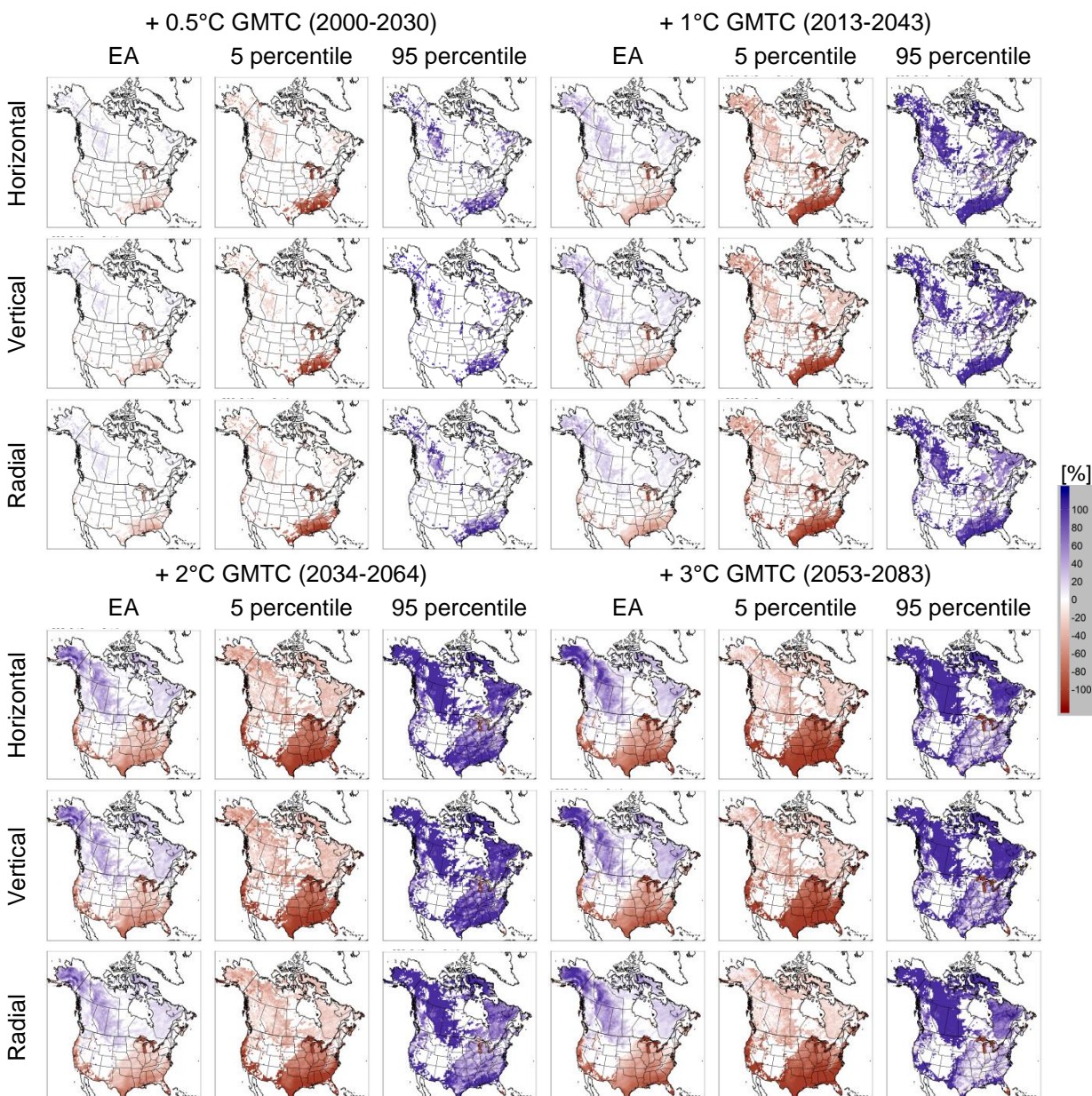

**Figure 5: Projected changes of CanRCM4 (ensemble average (EA) and 5 & 95 percentiles) to the same design ice thickness for +0.5, +1, +2, and +3 °C GMTC levels, which are 2000-2030, 2013-2043, 2034-2064, and 2053-2083 periods based on CanESM2 RCP8.5 scenario, respectively, with respect to the baseline period. Projected changes are marked by colors when baseline and future ensembles are statistically significant with two-sample t-test at the 10% significance level.**

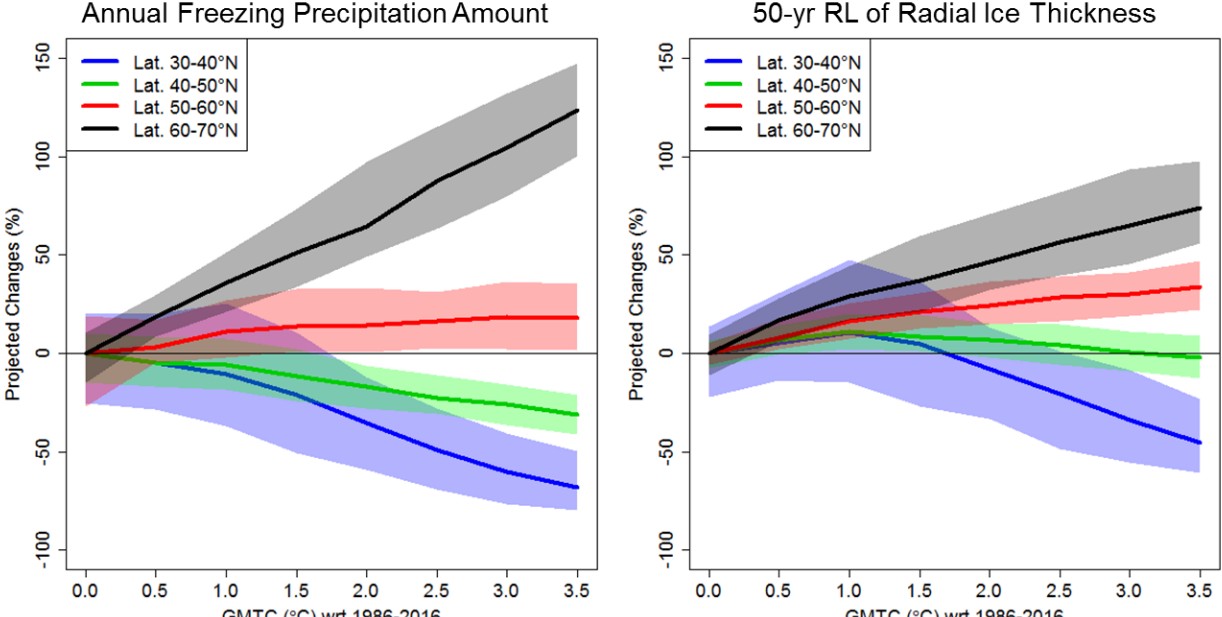

**Figure 6: Zonal averages of projected changes in CanRCM4 large ensemble to annual FP amount (left panel) and 50-year RL radial ice thickness (right panel) for four different latitude zones (i.e., 30-40°N, 40-50°N, 50-60°N and 60-70°N latitudes) over North America as a function of global mean temperature change (GMTC) relative to the baseline period. Minimum and maximum values of areas and lines represent 5 & 95 percentiles and median values, respectively, obtained from the ensemble.**

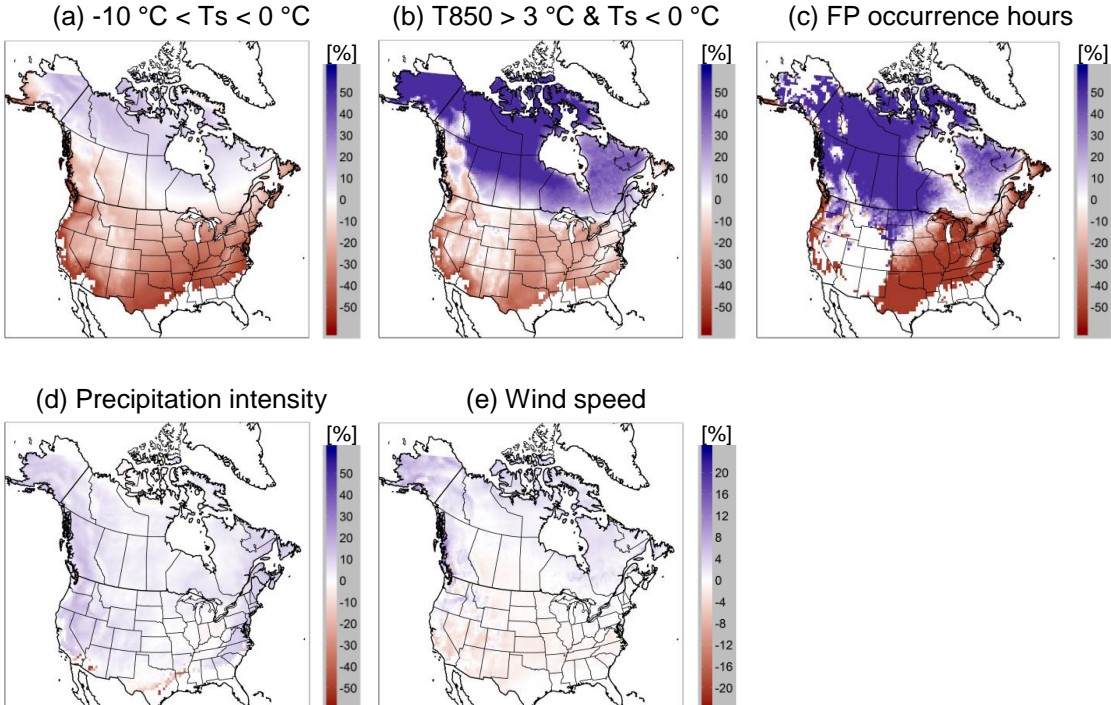

**Figure 7: Projected changes to frequencies of (a) surface temperature (Ts) condition (i.e., -10 °C < Ts < 0 °C) and (b) combined Ts and temperature at 850 hPa (T850) condition (i.e., T850 > 3 °C & Ts < 0 °C), representing favorable temperature conditions for FP occurrence, as well as (c) freezing precipitation (FP) occurrence hours from CanRCM4 large ensemble at +2°C GMTC level with respect to the baseline period. Projected changes to precipitation intensity and surface wind speed from the ensemble at the same GMTC level are presented in (d) and (e). All values are calculated for the freezing season (i.e., from onset to offset of Ts < 0°C) for each year.**

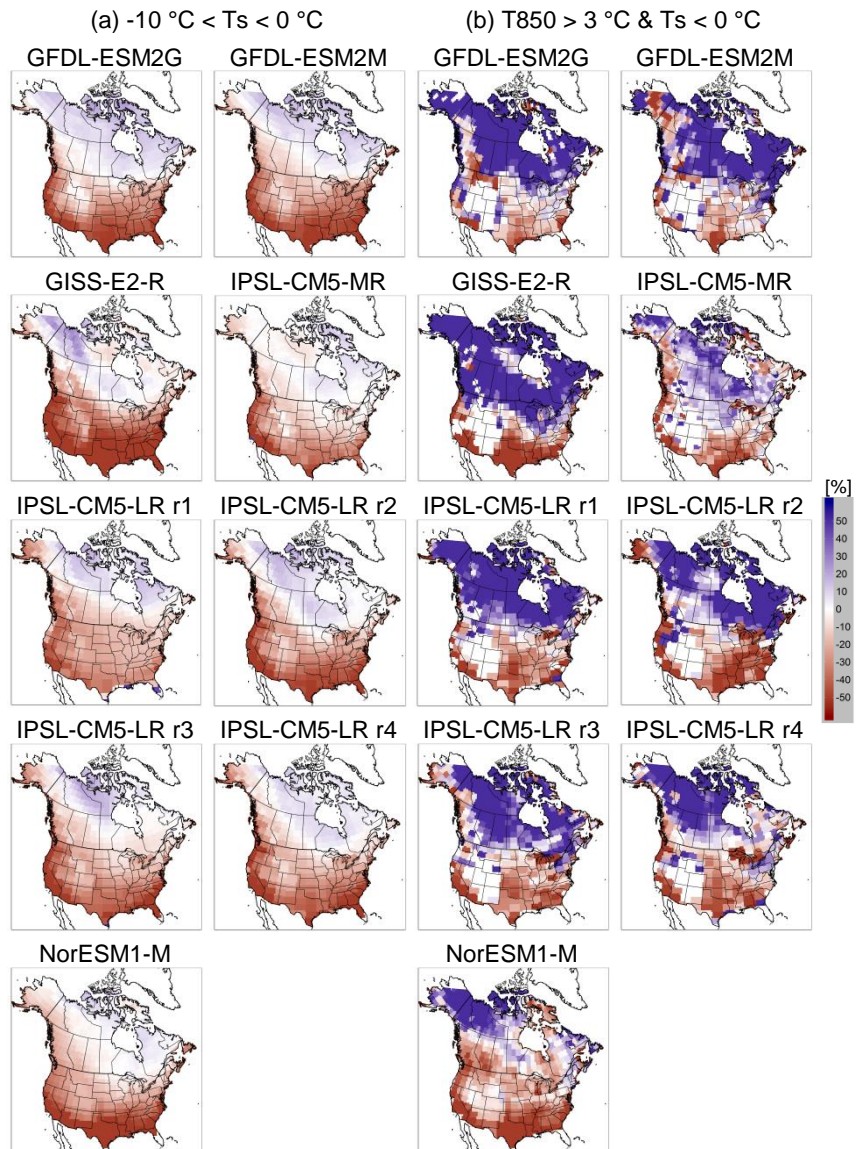

**Figure 8: Projected changes to frequencies of (a) -10 °C < Ts < 0 °C and (b) T850 > 3 °C & Ts < 0 °C from 9 GCMs at their +2°C GMTC level relative to the baseline period, respectively.**