# Peer review of "Projected changes to extreme freezing precipitation and design ice loads over North America based on a large ensemble of Canadian regional climate model simulations"

_Natural Hazards and Earth System Sciences, 2018_

## Referee Comment (RC1) · Mirela-Adriana Anghelache (Referee) · 7 Mar 2019

This scientific article is very well structured and shows an exhaustive work on its subject. There are many abbreviations due the overviews and comparisons of several simulation models. Yet, there are some left aside or explained at the bibliography. That's why I suggest that some abbreviations to be explained inside the text or better in a footnote, for example the term CSAI. Actually, I think that the insertions of footnotes is a good option, that shortens the phrase and give the reader the opportunity to see immediately the meanings of uppercase letters. In the sub chapter, 2.3 GCM simulations, there is an enumeration which reader is tempted to skip because of its lengths. Maybe it's better to be reformulated, broken in smaller sentences, just to capture the reader's attention, too. My congratulations to the authors for their research.

---

## Referee Comment (RC2) · Anonymous Referee #2 · 7 Mar 2019

I have to congratulate the authors for their excellent work. I have only a minor suggestion which will not impact the quality of the manuscript if not considered important by the authors. I think it would be very nice to add some words about the reason to select RCP 8.5. While speaking about freezing, one can expect that such phenomena will be less frequent in a warmer world.

---

## Author Comment (AC1) · 14 Mar 2019

We thank Mirela-Adriana Anghelache for these constructive comments to improve the readability of this manuscript. The abbreviations have been expended in the text of the revised manuscript. Please see lines 1, 8, and 13 on page 2. Although, we agree that footnotes can be a better way to explain the abbreviations for readers, we have avoided them based on the suggestion of the NHESS. Chapter 2.3 has been reformulated in the revised manuscript for better understanding, based on this comment. Please see lines 5-13 on page 5.

---

## Author Comment (AC2) · 14 Mar 2019

We thank anonymous referee for reviewing and providing comments on this manuscript. RCP8.5 was selected as it is an appropriate scenario for business-as-usual and non-climate policy conditions, which has closely matched recent decade emissions. This has been addressed in the revised manuscript. Please see lines 7-10 on page 4. There is a confidence that future freezing precipitations and associated atmospheric ice accretions will be less frequent in a future warming climate, particularly over most of mid (< 50°) latitude regions, as they generally occur when surface

temperatures are near freezing point (e.g., -10 $\sim$ 0°C). However, increases in those frequencies are also projected over high latitude regions due to increases in future upper level and surface temperatures and associated poleward shift of the surface 0 °C isotherm. Therefore, the changes in the freezing frequency are highly dependent on global warming levels and associated increases in upper level and surface temperatures at each region as shown in Sections 4.3 and 4.4.

---

## Author Comment (AC3) · 14 Mar 2019

**Projected changes to extreme freezing precipitation and design ice loads over North America based on a large ensemble of Canadian regional climate model simulations**

Dae Il Jeong[1], Alex J. Cannon[2], Xuebin Zhang[1]

[1] Climate Research Division, Environment and Climate Change Canada, Toronto, Ontario, M3H 5T4, Canada
[2] Climate Research Division, Environment and Climate Change Canada, Victoria, British Columbia, V8W 2Y2, Canada

*Correspondence to*: Dae Il Jeong (daeil.jeong@canada.ca)

**Abstract.** Atmospheric ice accretion caused by freezing precipitation (FP) can lead to severe damage and failure of buildings and infrastructure. This study investigates projected changes to extreme ice loads – those used to design infrastructure over North America (NA) – for future periods of specified global mean temperature change (GMTC), relative to a recent 1986-2016 period, using a large 50 member initial condition ensemble of the CanRCM4 regional climate model driven by CanESM2 under the RCP8.5 scenario. The analysis is based on three-hourly ice accretions on horizontal, vertical, and radial surfaces calculated based on FP diagnosed by the offline Bourgouin algorithm as well as wind speed during FP. The CanRCM4 ensemble projects an increase in future design ice loads for most of northern NA and decreases for most of southern NA and some northeastern coastal regions. These changes are mainly caused by regional increases in future upper level and surface temperatures associated with global warming. Projected changes in design ice thickness are also affected by changes in future precipitation intensity and surface wind speed. Changes in upper level and surface temperature conditions for FP occurrence in CanRCM4 are in broad agreement with those from nine global climate models, but display regional differences under the same level of global warming, indicating that a larger multi-model, multi-scenario ensemble may be needed to better account for additional sources of structural and scenario uncertainty. Increases in ice accretion for latitudes higher than 40°N are substantial and would have clear implications for future building and infrastructure design.

**1 Introduction**

Atmospheric ice accretion caused by ice storms is a frequent and major natural disturbance that can affect wind energy generation (Yang et al., 2015), urban functioning (Armenakis and Nirupama, 2014), communication structures (Mulherin, 1998), the forestry sector (Proulx and Greene, 2001), and electrical infrastructure (Rezaei et al., 2016; Jeong et al., 2018). In particular, extreme ice loads can result in severe damage, including the possibility of failure, to buildings and infrastructure such as bridges, antennas, towers, overhead transmission lines, and other structures. Therefore, 
[revised manuscript text omitted]

---

## Author Comment (AC4) · 14 Mar 2019

Revised manuscript

Please also note the supplement to this comment:
https://www.nat-hazards-earth-syst-sci-discuss.net/nhess-2018-395/nhess-2018-395-AC4-supplement.pdf
* * *
[Figure]

2018-395, 2018.

---

## Author Response (AR1)

**Reply to Professor Maria Bostenaru Dan (Editor)**

**Comments:** Congratulations for a well designed research. Thank you for the submission to this issue. The hazard described is relevant to know more about, in this issue and in the journal. The abstract summarises the essential, and the structure of the paper is as it should be. Given that it was extremely hard to find referees, the relevance of the paper is not underlined enough. The introduction part can be improved. Thus, apart that there are relatively many self references, I suggest including, according also to the anonymous referee, some references, and pictures when available, on the effects of freezing rain. For example we had in Romania freezing rain in 2019 (see attached file, but for Canada much more relevant are the electricity lines) at the End of January. It destroyed much of the green spaces. For Canada, I suggest writing on the ice storm of 1998, and how it affected the heat supply, which there relies much on electricity, given that there is much hydropower. This would also underline the importance for infrastructure. It would be helpful to localise the areas of freezing rain, connected to this infrastructure and the hydropower, as well as to mention other events than 1998, which might exemplify the relevance for Canada.

**Answer:** We thank Professor Maria Bostenaru Dan for the valuable comment to improve the introduction by addressing natural hazards caused by ice storms over Canada. We also thank her for the pictures showing damages in Romania by the ice storm occurred in 2019. As suggested, specific damages caused by the ice storm occurred in 1998 over eastern Canada and the northeastern United States (Lecomte et al., 1998; Cortinas, 2000; Hauer et al., 2011) as well as the ice storm occurred in 2013 over eastern Canada (Armenakis and Nirupama, 2014; City Report, 2014) have been addressed in the revised manuscript. Please see from page 1, line 26 to page 2, line 5. It is a good idea to provide some pictures showing the damages of infrastructure by the two ice storms; however, they have not been included, as we are worried about copyright issues and people can easily find those pictures through internet. Several studies have been additionally introduced in the fields of urban functioning (Hauer et al., 2011), the forestry sector (Seidl et al., 2017), and electrical infrastructure (Fu et al., 2006) (lines 23-26 on page 1). We hope that the revised manuscript adequately addresses the editor's suggestion.

**References**

Armenakis, C. and Nirupama, N.: Urban impacts of ice storms: Toronto December 2013, Nat. Hazards, 74(2), 1291-1298, 2014.

Fu, P., Farzaneh, M., and Bouchard, G.: Two-dimensional modelling of the ice accretion process on transmission line wires and conductors, Cold Regions Science and Technology, 46(2), 132-146, 2006.

Cortinas J.: A climatology of freezing rain in the Great Lakes region of North America, Monthly weather review, 128(10), 3574-3588, 2000.

Hauer, R.J., Hauer, A.J., Hartel, D.R., and Johnson, J.R.: Rapid assessment of tree debris following urban forest ice storms, Arboriculture and Urban Forestry, 37(5), 236, 2011.

Henson, W., Stewart, R., and Kochtubajda, B.: On the precipitation and related features of the 1998 ice storm in the Montréal area, Atmospheric Research, 83(1), 36-54, 2007.

Lecomte, E.L., Pang, A.W., and Russell, J.W.: Ice storm '98', Ottawa, Canada: Institute for Catastrophic Loss Reduction, p. 99, 1998.

Seidl, R., et al.: Forest disturbances under climate change, Nature climate change, 7(6), 395, 2017.

---

## Author Response (AR2)

**Reply to Professor Maria Bostenaru Dan (Editor)**

**Comments:** Thank you for addressing my suggestions. I agree that if there are no own photos of the damage, it is complicated to acquire publication rights of photographs, since they also count as art works. I appreciate that the author addresses in the author's response that the damages cased by freezing rain express the impact of natural hazards on urban areas and infrastructure. This should be explicitly addressed in a sentence in the revised section, highlighting the relevance for the special issue.

**Reply:** We thank Professor Maria Bostenaru Dan to understand our concern about copyright issues. We also agree that the introduction has been significantly improved by highlighting the relevance between this study and the special issue. We thank for the comment again.